# Brief Communication: From modelling to reality - Flood modelling gaps highlighted by a recent severe storm surge event along the German Baltic Sea coast

Joshua Kiesel[1*]; Claudia Wolff[2*]; Marvin Lorenz[3]

[1]Institute for Environmental Studies (IVM), Vrije Universiteit Amsterdam, Amsterdam, 1081HV, The Netherlands
[2]Institute of Geography, Kiel University, Kiel, 24118, Germany
[3]Leibniz Institute for Baltic Sea Research Warnemünde, Rostock, 18119, Germany

*Correspondence to*: Claudia Wolff (wolff@geographie.uni-kiel.de) and Joshua Kiesel (j.kiesel@vu.nl)

**Abstract.** In October 2023, Germany and Denmark's Baltic Sea coasts experienced a severe storm surge, predominantly impacting the German state of Schleswig-Holstein and parts of southern Denmark. The surge led to extensive flooding in cities like Flensburg and Schleswig, causing the breaching of at least six (regional) dikes and causing over 200 million Euros in damages in Schleswig-Holstein. By chance, the peak water levels of this storm surge aligned well with those of recent hydrodynamic flood modelling studies of the region. This rare coincidence offers crucial insights for our understanding of flooding impacts, flood management and modelling. By comparing those studies to the real-world example using extensive media reports, we aim to extract key insights and identify gaps to be tackled in order to improve flood risk modelling in the Baltic Sea region and beyond.

## 1 Introduction

The Baltic Sea is a semi-enclosed, microtidal marginal sea in the eastern North Atlantic, which is under pressure from a multitude of anthropogenic disturbances and natural hazards (Rutgersson et al., 2022; Reusch et al., 2018). On October 20th - 21st, an exceptional storm surge inundated parts of the German and Danish Baltic Sea coasts, demonstrating why both states are projected to experience the largest absolute coastal flood damage in Europe over the course of the 21st century (Rutgersson et al., 2022; Vousdoukas et al., 2020). Most affected during the October 2023 surge were the German federal state of Schleswig-Holstein and southern Denmark. This surge highlighted the extent of damages that can occur from events within anticipated coastal protection design parameters, as it led to extensive flooding in major cities such as Flensburg, Schleswig and Eckernförde (all of which located in the German federal state of Schleswig-Holstein, Fig. 1), breaching a minimum of six (regional) dikes (NDR, 2024d) and causing preliminary damages of up to 200 million Euros in Schleswig-Holstein alone (NDR, 2024e).

The October 2023 surge, which was extensively covered in the media and similar in magnitude as recent studies from the same region (Höffken et al., 2020; Kiesel et al., 2023b; Kupfer et al., 2024), poses a unique opportunity to compare flood modelling

studies with a real event and thereby reflecting on flood modelling capabilities and existing gaps. This enables deriving critical insights that may assist in developing modelling strategies for the Baltic Sea coast and beyond.

## 1.1 Characterisation of the event

The storm of October 20, 2023, driven by strong easterly winds, persisted for two days and reached peak wind speeds of 102 km/h. The primary cause of this event was the air pressure difference between a high-pressure system over Scandinavia (1030 hPa) and a low-pressure system over England (975 hPa), resulting in a strong and sustained easterly wind field across the entire Baltic Sea (Bundesamt für Seeschifffahrt und Hydrographie, 2024). Strong easterly winds, as experienced during the October surge, constitute the primary cause of storm surges along the German Baltic Sea coast. Under such conditions, the German federal state of Schleswig-Holstein (SH) is exposed to a longer fetch length, explaining why extreme sea levels are typically higher than in the state of Mecklenburg-Western Pomerania (MP) (Gräwe and Burchard, 2012; Kiesel et al., 2023b).

Days before the October 2023 surge, water levels in the Kiel- and Lübeck Bay were already 20–50 cm above mean sea level, which is referred to as "preconditioning" (Bundesamt für Seeschifffahrt und Hydrographie, 2024). Preconditioning describes elevated water levels within the Baltic Sea before the onset of a storm, which is an important factor in the development of extreme water levels (Weisse et al., 2021).

Across the German Baltic Sea coast, the October 2023 surge caused the highest peak water levels in Flensburg (Table 1, Figure 1). In this city, only the storm surge of November 13, 1872, was higher than the October 2023 flood, making this recent surge the second-highest on record in the past 150 years (Bundesamt für Seeschifffahrt und Hydrographie, 2024). In Flensburg, water levels remained over 1.0 m above mean sea level for 53 hours and over 2.0 m for 9 hours. At several tide gauges, the October 2023 storm surge was roughly equivalent to a 200-year event, as calculated from a hindcast of a hydrodynamic model of the western Baltic Sea covering the years 1961 - 2018 (Kiesel et al. 2023b; please see Table 1). We note, however, that extrapolating extreme sea levels beyond the length of tide gauge records is sensitive to the length of the data included. For instance, McPherson et al., (2023) could show that extreme sea levels extrapolated using the limited time series of available tide gauge records along the German Baltic Sea coast can be underestimated.

**Table 1: Observed peak water levels during the October 2023 surge at 21 selected tide gauges along the German Baltic Sea coast. Observational data were taken from EMODnet (2020). The return water levels of a 200-year storm surge were taken from Kiesel et al. (2023b). In their study, the authors have used a hindcast simulation of a western Baltic Sea hydrodynamic model between the years 1961 and 2018 to extrapolate the extreme sea levels. An asterisk next to the tide gauge location denotes that the observed water level of the October 2023 surge ranges within the error margin of a 200-year event as calculated by Kiesel et al., 2023b. Note that the modelled 200-year return surges are detrended for sea-level rise. Values written in bold and italic letters indicate that tide gauge data were used for extreme value extrapolation instead of the hydrodynamic model. Note that the tide gauges of Kappeln and Schleswig stopped working a while before the peak of the October 2023 surge reached their locations.**

| No | Station | Date | Time (MEZ) | Observed water level relative to NN (cm) | 200-year return water level (cm) |
|----|---------|------|------------|------------------------------------------|----------------------------------|

| | | | | | |
|---|---|---|---|---|---|
| 1 | Althagen* | 21-10-2023 | 12:42:00 | 100 | 110±36 |
| 2 | Barhoeft* | 20-10-2023 | 21:38:00 | 143 | 153±12 |
| 3 | Eckernfoerde* | 20-10-2023 | 21:10:00 | 215 | 205±28 |
| 4 | Flensburg* | 20-10-2023 | 22:40:00 | 227 | 203±29 |
| 5 | GreifswalderOie* | 20-10-2023 | 18:53:00 | 148 | 168±23 |
| 6 | Heiligenhafen* | 20-10-2023 | 23:07:00 | 172 | 190±35 |
| 7 | Kappeln* | 20-10-2023 | 13:00:00 | 163 | 151±23 |
| 8 | KielHoltenau* | 20-10-2023 | 21:33:00 | 195 | 203±29 |
| 9 | Koserow | 20-10-2023 | 19:05:00 | 108 | 156±26 |
| 10 | Langballigau* | 20-10-2023 | 22:16:00 | 221 | 199±28 |
| 11 | Kalkgrund Leuchtturm* | 20-10-2023 | 22:57:00 | 208 | 196±27 |
| 12 | Neustadt* | 20-10-2023 | 18:38:00 | 180 | 199±33 |
| 13 | Rostock* | 20-10-2023 | 22:15:00 | 150 | 175±31 |
| 14 | Sassnitz | 20-10-2023 | 22:04:00 | 114 | 144±13 |
| 15 | Schleimuende* | 20-10-2023 | 21:32:00 | 208 | 198±26 |
| 16 | Schleswig | 20-10-2023 | 07:35:00 | 176 | 148±21 |
| 17 | Stralsund* | 20-10-2023 | 19:28:00 | 151 | 158±21 |
| 18 | Timmendorf* | 20-10-2023 | 18:50:00 | 161 | 194±38 |
| 19 | Ueckermuende* | 20-10-2023 | 17:44:00 | 92 | 111±30 |
| 20 | Warnemuende* | 20-10-2023 | 22:37:00 | 148 | 174±32 |
| 21 | Wismar* | 20-10-2023 | 22:20:00 | 158 | 197±40 |


Although peak water levels during the October 2023 storm surge were mostly below the design water level for state dikes along many coastal sections (200-year event + wave overflow + buffer for SLR) (Ministerium für Energiewende, Landwirtschaft, Umwelt, Natur und Digitalisierung des Landes Schleswig-Holstein, 2022), the event caused widespread and costly damages, including dike failures and flooding. Dike failures, however, were only observed along regional dikes. In

contrast to state dikes, regional dikes are in the responsibility of the water and soil associations, and are built according to variable and generally lower design heights (Hofstede, 2024). The total length of regional dikes in Schleswig-Holstein is 40.1 km - half of which did not experience damage from the storm surge, while about a third sustained medium (5.3 km) and severe (6.7 km) damage (Oelerich, 2024).

In the aftermath of the event, the first estimates of damages in the federal state of Schleswig-Holstein alone sum up to 200 million euros, of which around €40 million are associated with coastal protection and €140 million with touristic and municipal infrastructure (NDR, 2024e). Examples include a dike breach and the drowning of livestock near Damp, Schleswig-Holstein. The total damage costs in this area were estimated at ten million Euros (SHZ, 2024). Wieck am Darß (located in the federal state of Mecklenburg Western Pomerania) experienced two dike breaches on a total length of 30 m, posing a flood threat to 75

houses (Nordkurier, 2024). The Baltic Sea channel called The Schlei and adjacent open coasts were particularly impacted. In this region, three regional dikes breached in Arnis, Maasholm and south of the harbour of Olpenitz (NDR, 2024c). In Arnis, temporary repairs were carried out with over 30,000 sandbags, and full repairs are scheduled for spring (Kieler Nachrichten, 2024). Schleimünde and the Lotseninsel, both located on a large barrier spit system that marks the Schlei's inlet, experienced significant damage to coastal protection infrastructure, elevating the risk of further damage (NDR, 2024b). Another dike breach

happened north of Falshöft (Geltinger Birk, Flensburg Fjord, Schleswig-Holstein), where the dike collapsed on a length of approximately 600 m, which led to extensive flooding. Due to years of embankment, vast parts of the area behind the breached dike are below sea level. It therefore took a week to pump the water out of the area, and damages of €3-5 million contributed to the overall financial toll (NDR, 2024f).

The October 2023 storm surge has also demonstrated the effectiveness of natural buffer zones between the dikes and the sea. During that surge, dikes that were located further inland behind natural buffers such as beach ridges were not damaged, while strong damages were observed along dikes that are located directly behind the beach (Hofstede, 2024). The potential effectiveness of a natural buffer zone is further demonstrated by the example of the dike breach at Geltinger Birk, located in the east of Flensburg fjord. Even though the dike breach was approximately 600 meters wide, this failure did not lead to

damaged buildings, as the area behind the breached dike is part of a large-scale wetland and lagoon restoration scheme. In 2013, a controlled rewetting of the area was initiated by raising water levels by 2.5 m on an area of about 1000 ha. At the same time, a new ring dike was constructed, now providing effective coastal protection for the adjacent village of Falshöft – also during the nearby dike breach of the October 2023 surge (Schernewski et al., 2018). The availability of potential areas for implementing such buffer zones along the German Baltic Sea coast through managed realignment and their potential to mitigate

the impacts of storm surges has recently been demonstrated by Kiesel et al. (2023a), The experiences from the October 2023 surge and the demonstrated potential and effectiveness of large-scale buffer zones along the German Baltic Sea coast for mitigating coastal flooding strongly support the case for moving (and/or constructing) flood barriers further inland wherever possible.

**2 Flood modelling gaps highlighted by the October 2023 surge**

**2.1 Emphasizing hydrograph variability and spatial dependencies in coastal flood modelling**

Current methodologies for assessing coastal flooding along the German Baltic Sea coast typically employ a location specific design surge with a uniform return period across different regions, as exemplified by Kiesel et al., (2023a, b). Such location specific design surges are furthermore used to determine the design height of coastal protection measures (e.g. the 200-year

event as used for state dikes along the German Baltic Sea coast), which ensures a common protection standard for all people

across a region. Therefore, using a regionally uniform return period to assess the impacts or effectiveness of dikes on todays and future coastal flooding and exposure of populations constitutes a meaningful approach. However, the latter approach neglects the unique spatial characteristics and dependencies of extreme events, often referred to as spatial footprint or spatial dependence (Enríquez et al., 2020; Li et al., 2023), which basically describe the fact that such extremes are unlikely to happen

simultaneously across the entire region. The significance of spatial dependence was evident during the October 2023 surge along the German Baltic Sea coast. This event, driven by strong easterly winds, which have a longer fetch length for the German federal state of Schleswig-Holstein than for Mecklenburg-Western Pomerania, resulted in varying impacts across the region. Schleswig-Holstein experienced peak water levels surpassing the simulated 200-year return levels, unlike Mecklenburg-Western Pomerania. This disparity, illustrated in Figure 1, underscores the necessity of considering spatial

dependence for accurate regional and particularly transnational risk assessments and damage estimations, particularly in the light of disaster management measures and compensation funds (Jongman et al., 2014).

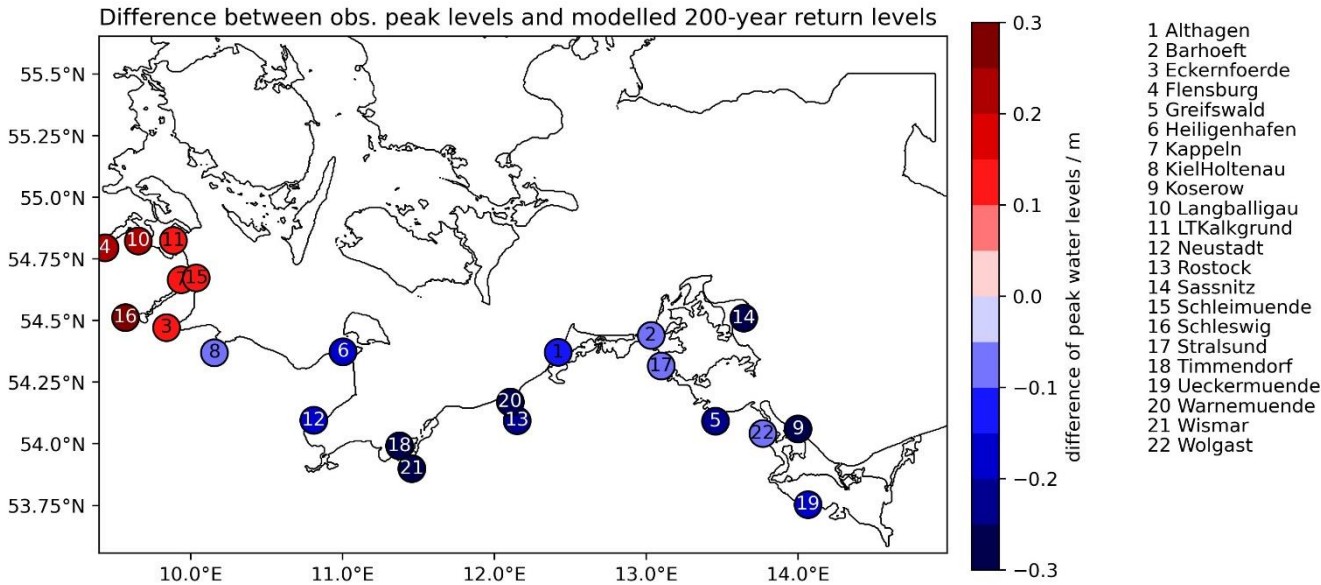

**Figure 1: Map with locations of tide gauge stations from Table 1. The colours depict the difference in peak water levels between the observed October 2023 storm surge and the modelled 200-year return water levels from Kiesel et al. 2023b. Red colours indicate that extrapolated 200-year return water levels were lower than the observed peaks of the October 2023 surge and blue colours show that the latter was lower than the constructed 200-year events.**

Further comparisons with the analyses of Kiesel et al. (2023b) showed that while the peak of the 200-year return water levels by chance broadly matched the October 2023 surge, discrepancies existed in the temporal evolution of the synthetic surges

compared to the real-world example (Figure 2, Table 1). Excluding the influences of short surface waves, storm tide hydrographs are a function of mean sea level, astronomical tide and storm surge (Pugh 1996; Lewis et al., 2011). Tides can be excluded as a cause for the observed differences in hydrographs since the Baltic Sea is characterized by a microtidal regime.

In addition, short surface waves were not considered in Kiesel et al., (2023b). Differences in mean sea level might have affected the hydrographs only in terms of peak water levels, considering that the simulated 200-year design surges were detrended for sea level rise. Consequently, the differences in the shapes of the hydrographs can only originate from differences in storm surge characteristics.

The constructed design hydrographs were derived from a coastal ocean model covering the western Baltic Sea, using hindcast model runs (1961-2018) for each location depicted in Figure 1. Only surges with peak water levels higher than 1 meter above mean sea level were taken into account. Ultimately, the remaining surges were averaged in their temporal evolution (Kiesel et al., 2023b).

While the constructed surge hydrographs of Kiesel et al. (2023b) align well with the October 2023 observations within protected lagoons of Mecklenburg-Western Pomerania (Figure 2j, k, m, n,o), locations at the open coast show differences in the onset of event. The rise of water levels during the actual storm surge of October 2023 was mostly slower than the modelled events (Figure 2a-i). This reveals an underestimation of surge duration in the constructed hydrographs at the open coast. Recent studies for the Baltic Sea cities of Lübeck and Eckernförde have demonstrated that longer surge durations with identical peak water levels can result in larger flooding extents, with variances of up to 60 % (Höffken et al., 2020; Kupfer et al., 2024). These studies suggest that a comparison of flood extents between the October 2023 surge and the synthetic 200-year events from Kiesel et al., (2023b) would likely reveal an underestimation in the model simulations. This has implications for coastal management, as stakeholders with a very low tolerance to uncertainty may require high-end sea-level rise scenarios (Hinkel et al., 2019), thus are likely in need of knowledge regarding high-end flood risk estimates in order to prepare for the worst case. Therefore, it is crucial to assess the sensitivity of flooding extents to hydrograph shapes/surge durations, as results can be highly case-specific. Given the computational demands of more nuanced probabilistic assessments (e.g. Kupfer et al. 2024) and the practical limitations of available resources, a focus on surge shapes associated with longer (upper percentile) durations can offer a pragmatic solution for the analysis of coastal flooding caused by rare and impactful events.

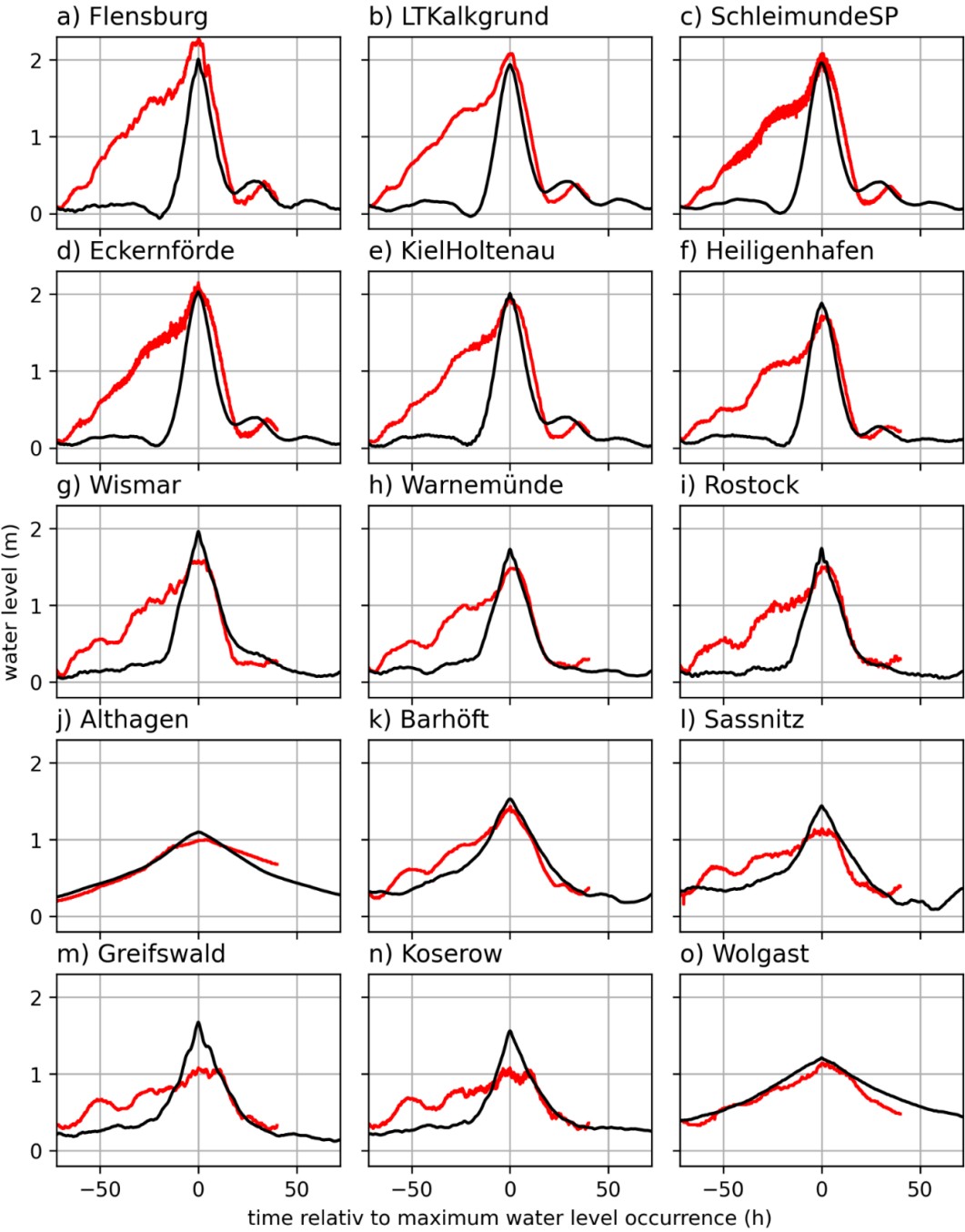

**Figure 2:** **Comparison of hydrographs of the observed October 2023 surge (red) and the constructed 200-year return level events of Kiesel et al. (2023b) (black) for some of the stations listed in Table 1. Note that the constructed hydrographs do not consider mean sea level offsets.**

## 2.2 Neglected Factors: Dike breaches and morphodynamic responses in flooding extent and depth estimations

The October 2023 surge has highlighted critical gaps in current broad-scale flood modelling, particularly regarding dike breaches and morphodynamic responses of the shoreline. In flood risk research, fragility curves are used to assess the probability of dike (or dune) failure as a consequence of a specific hydraulic loading (Vorogushyn et al., 2009). However, site specific fragility curves depend on detailed information about flood defense structure and foundation properties (Simm et al., 2009). In addition, using two-dimensional hydrodynamic models, which are currently state-of-the-art in flood risk assessments, can be impractical when multiple breach and loading scenarios are simulated on large spatial scales. Among other reasons, the latter is due to high computational costs (Simm et al., 2009). This reasoning might partly explain why the wider implementation of fragility curves as a probabilistic framework for studying the impacts of dike failure in broad-scale hydrodynamic flood modelling studies is currently limited. However, in most cases, the primary reason is the absence or inconsistency of geospatial data on coastal protection infrastructure (Hinkel et al., 2021; Vousdoukas et al., 2018b).

Beyond that, the October 2023 surge underscores that even when data on coastal protection infrastructure is available, neglecting the potential for dike breaches can lead to underestimations of flooding extent and associated damages. The above becomes evident when comparing the regional modeling output of Kiesel et al. (2023b) with the October 2023 surge. While Kiesel et al. (2023b) included the location and height of natural and anthropogenic coastal protection structures, only one of the six dikes that reportedly breached during the October 2023 surge (Wieck auf dem Darß, Mecklenburg Western Pomerania) was simulated to overflow during the constructed 200-year event. This overlooks the possibility of dikes breaching even before water levels reach the crest height (Bomers et al., 2019). Several dike breaches during the October 2023 surge demonstrate how this can lead to underestimations in flooding extent and associated damages (see section 1.1).

The observed dike breaches that could not be accounted for in Kiesel et al., (2023b) demonstrate that current flood modeling needs to strengthen efforts to incorporate the possibility of dike failure in broad-scale assessments. This unresolved knowledge gap has to do with missing data on the location, design height, building material and current condition of dikes, and the limited process understanding due to the highly stochastic nature of breaching, which would require high-resolution hydromorphodynamic modelling (Hinkel et al., 2021; Vousdoukas et al., 2018a). Ways forward ultimately depend on the availability of dike specific data. For instance, a ductile dike behavior can result in limited water volumes flowing through established breaches, which is dependent on building material (den Heijer and Kok, 2023). Once such data is available, existing probabilistic approaches (e.g. dike fragility curves) can be expanded, such as the one introduced by Vorogushyn et al. (2010), that uses hydraulic loads and dike resistance to assess dike fragility. Without making use of high-resolution and thus computationally expensive online-coupled hydromorphodynamic models, locations along the coast could be identified, where dike breaches are most likely to happen. Once a site-specific critical hydraulic load is reached, the model could be re-run

assuming a breach at that specific location. Certainly, rerunning broad-scale models is ultimately dependent on computational
resources, which is particularly true once uncertainty bounds for dike fragility are included.

Additionally, neglecting the morphodynamic response of natural flood barriers like dunes and beach ridges to extreme water
levels and waves can lead to underestimations in flooding extent and damages (Toimil et al., 2023). Along the German Baltic
Sea coast, beaches and dunes are widely acknowledged for their coastal protection function, which is why they are maintained
by means of sand nourishment (Tiede et al., 2023). Sand nourishment aims to stabilize the shoreline position and maintain
dune width and height, reducing the risk of collapse during an extreme event (Claudino-Sales et al., 2008). However, sand
nourishments are a costly endeavor. For instance, of the annual 15.5 million Euro that Mecklenburg-Western Pomerania has
spent between 1990 and 2008 on coastal protection measures, 45.6 % are spent on  nourishments (Staatliches Amt für Umwelt
und Natur Rostock, 2021; Tiede et al., 2023). More frequent and more intense storm surges may increase the rhythm of such
nourishments (Vousdoukas et al., 2017). Along the Baltic Sea coast of the German federal state of Mecklenburg-Western-
Pomerania, beach nourishments are currently taking place every 5-10 years (Tiede et al., 2023). In Ahrenshoop (Mecklenburg-
Western Pomerania), a major nourishment took place in 2021, but the October 2023 surge washed away parts of the beach,
leaving the adjacent dunes exposed to further erosion (NDR, 2024g). In Schönberg (Schleswig Holstein), where the October
2023 storm eroded 30,000 m³ of sand, the need for new nourishments may produce costs of up to 1.5 million Euros (NDR,
2024a). Thus, neglecting morphodynamic processes such as shoreline erosion when simulating coastal flooding and associated
damages can lead to underestimated damage costs. Furthermore, the reduced width of beaches and dunes increases the risk of
dune collapse during subsequent storm surges, even if the second storm surge is not of the same magnitude as the first.

**2.3 Secondary event dynamics and their unexplored potential for amplified damages (cascading events)**

The study of consecutive events is part of the multi-hazard research domain and one of four categories describing the
interrelations of multi-hazards (Claassen et al., 2023). Consecutive events are a critical yet often under-researched domain.
They can cause more extensive damage than would be the case when they occurred in isolation, making this a high priority
research area in the field of natural hazards (De Ruiter et al., 2020). This highlights the importance of incorporating the
potential impacts of successive flooding events into assessments. Such modelling studies, which consider a sequence of flood
events along with the potential failure of coastal protection infrastructure, are vital for understanding the cumulative effects of
these incidents.

Erosion and dike breaches triggered by the October 2023 storm surge, for example, significantly increase the risk of the affected
coastal areas receiving subsequent flooding from consecutive events. For instance, the regular beach nourishments in
Ahrenshoop (Mecklenburg-Western Pomerania) have maintained shoreline and dune stability over the past decades (Tiede et
al., 2023), effectively providing a buffer against storm surges and erosion. However, parts of this buffer were washed away

during the October 2023 surge (NDR, 2024g), leaving the coast exposed to the potential impacts of a consecutive event. Similar problems may arise at locations where dikes have been breached and cannot immediately be repaired (Wieck auf dem Darß, Geltinger Birk, Arnis), or where dikes are considerably damaged thus delivering reduced coastal protection. The latter is exemplified by heavily damaged dikes in the aftermath of the October 2023 surge in eastern Schleswig-Holstein (NDR, 2024c).

In summary, the October 2023 event has left parts of the German Baltic Sea coast with substantially reduced natural and man-made coastal protection. This reduction exposes large areas to the impacts of consecutive surges, even those of lower magnitude. This and the fact that only those dikes got severely damaged that were located close enough to the shoreline (Hofstede 2024) may provide a strong argument for maintaining natural buffer zones between the sea and the developed land. Such buffer zones can benefit both ecosystems and humans. Since only approximately a third of the German Baltic Sea coast is protected by dikes, the planning of new constructions and coastal protection infrastructure that may become necessary in the future (Kiesel et al., 2023a,b) should take idea of buffer zones into account.

## 3. Towards an updated coastal flood research agenda

From the coincidence of recent studies on a regional scale and the occurrence of an extreme surge of similar magnitude, we derive insights and knowledge gaps in current coastal flood modeling. The October 2023 surge along the southwestern Baltic Sea coast has caused severe damage despite being within the design parameters of the state's coastal protection measures. Parts of the experienced damage can be explained by missing protection measures in locations where they can't be implemented, for instance due to lack of space. Among other causes, the latter is a consequence of the microtidal environment of the Baltic Sea, explaining why many settlements and infrastructure are located very close to the mean water line (Vafeidis et al., 2020). Such locations include densely populated and harbor areas, such as the cities of Eckernförde, Schleswig and Flensburg.

The October 2023 surge has also revealed a set of processes yet underrepresented in scientific studies. These currently widely disregarded processes clarify existing knowledge gaps that need to be addressed by the scientific community, as they may lead to substantial underestimations in flooding extent, depth and damages. These processes include; (1) the importance of hydrograph variability, which affects surge duration and flooding extent (see Fig. 2); (2) the incorporation of spatial dependencies when regional flood damages and risk are quantified; (3) morphodynamic feedback mechanisms such as the potential for dike breaches (or damages done to the dike without breaching), and; (4) consecutive events, where prior events can weaken coastal protection infrastructure, potentially leading to considerably increased flood damages of secondary events, even if these are of lower magnitude.

**Author contribution**

JK and CW conceptualized the scope and research aims of the study and prepared the original draft of the manuscript, with the support of ML. ML further contributed visualizations and data curation. All authors contributed to reviewing and editing
of the manuscript.

**Competing interests**

The authors declare that they have no conflict of interest.

**Acknowledgements**
The authors would like to thank Athanasios Vafeidis and Horst Sterr for fruitful discussions and for providing helpful feedback on the original draft of the manuscript. We also thank Annika Block for supporting us in compiling media reports and collecting information about the October 2023 surge. Furthermore, this work is a contribution to the ECAS-Baltic project: Strategies of ecosystem-friendly coastal protection and ecosystem-supporting coastal adaptation for the German Baltic Sea coast. The project was funded by the Federal Ministry of Education and Research (BMBF, funding code 03F0860H). We furthermore
acknowledge financial support by Land Schleswig-Holstein within the funding program Open Access Publikationsfonds.

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
