# Peer review of "Brief Communication: From modelling to reality - Flood modelling gaps highlighted by a recent severe storm surge event along the German Baltic Sea coast"

_Natural Hazards and Earth System Sciences, 2024_

## Author Comment (AC1)

**Brief Communication: From modeling to reality: Flood modelling gaps highlighted by a recent severe storm surge event along the German Baltic Sea coast**

**Authors response to the reviewers**

**In order to enhance readability, our responses are written in green and in-text citations are displayed in italics.**

**Reviewer 2:**

The authors have presented the challenges of coastal hazard and risk modeling through a case study of the storm surge levels and losses observed in one recent event, for the purpose of motivating future research agenda.

While the authors' analysis of the recent Oct 2023 event with respect to available hazard estimates is novel, their highlighted gaps have been extensively discussed in the broader disaster hazard, risk, and resilience literature. As a Brief Communication, the manuscript can be valuable to the scientific community and policymakers with some improvements.

We would like to thank R2 for the positive and helpful feedback. We agree with the reviewer that many aspects/limitations of flood modeling have been extensively discussed in the literature. However, our analysis is novel in its comparison of a hydrodynamic flood model forced with design hydrographs of statistically derived surge shapes and return water levels with a real-world event of similar magnitude. This rare coincidence enabled us to provide a timely and practical case study that underscores the relevance and limitations of current coastal flood models.

    1.   The title and the abstract were not clear and did not fully reflect the goals of the manuscript. "Insights from a [event]…" is very broad and could be narrowed down, for example, as "Modelling Gaps…". Similarly, the last statement of the abstract mentions proposing "strategies for improving flood risk modelling…", however, the manuscript appeared to focus mainly on highlighting gaps rather than proposing strategies for bridging those gaps.

We thank R2 for this remark and agree that the title and the scope of the manuscript as defined at the end of the abstract needs to be adjusted. In response, we have now adjusted the title ("**From modelling to reality – Flood modelling gaps highlighted by a recent severe storm surge event along the German Baltic Sea coast**") and have rewritten the abstract, which now reads:

*"In October 2023, Germany and Denmark's Baltic Sea coasts experienced a severe storm surge, predominantly impacting the German state of Schleswig-Holstein and parts of southern Denmark. The surge led to extensive flooding in cities like Flensburg and Schleswig, causing the breaching of at least*

*seven (regional) dikes and causing over 200 million Euros in damages in Schleswig-Holstein. By chance, the peak water levels of this storm surge aligned well with those of recent hydrodynamic flood modelling studies of the region. This rare coincidence offers crucial insights for our understanding of flooding impacts, flood management and modelling. By comparing those studies to the real-world example using extensive media reports, we aim to extract key insights and identify gaps to be tackled in order to improve flood risk modelling in the Baltic Sea region and beyond."*

2.  In Section 2.1, the authors highlight the differences between the 200-year hazard scenario with observations from the Oct event. In order for them to match better, the authors propose inclusion of spatial correlation, and using 60th percentile duration for the 200-year event. However, these modifications would not guarantee that the updated hazard would match any future event. This is commonly observed in hazard modelling as hazard only represents site-independent exceedance levels, and is necessarily different from individual scenarios over a wide geographic region.

Inclusion of spatial correlation can in fact help in providing better cumulative loss estimates, however, the authors did not include any comparisons between cumulative loss or damage risk estimates with the observed losses in order to support their proposals.

We thank R2 for this comment, which has helped us to understand better which points in the manuscript require further clarification. In agreement with R2's comment, we now better explain why we suggest to use higher percentile surge durations (i.e. storm surge hydrograph shapes) along with very extreme surges, such as a 200-year event, by referring to reported stakeholder needs (see new text in the revised version of the manuscript below).

*"This has implications for coastal management, as stakeholders with a very low tolerance to uncertainty may require high-end sea-level rise scenarios* (Hinkel et al., 2019), *thus are likely in need of knowledge regarding high-end flood risk estimates in order to prepare for the worst case."*

First, we would like to point out that our study is not aiming at providing a better flood forecast. We are comparing a regional scale hydrodynamic modeling study with a real-world extreme event, as both were by chance similar in terms of peak water level (as now added to the abstract and the introduction). In the modelling study, the 200-year return water levels had to be extrapolated and synthetic hydrographs were extracted, as the publicly available data of many tide gauges in the study region had not seen an event of such magnitude until the original paper was published. However, as the October 23 surge happened, we found that differences were evident in the shape of the observed and synthetic hydrographs. The October 23 surge turned out to be longer in duration and we know from previous studies from the German Baltic Sea coast that longer surge durations can lead to larger flood extents (Höffken et al., 2020; Kupfer et al., 2024). Thus, when we could compare the simulated flooding extents from Kiesel et al. (2023) with those of the October 23 surge, we would most likely find that the October 2023 surge resulted in a larger flood extent, even though both events were of similar magnitude in terms of peak water level.

Stakeholders and decision makers with a very low tolerance to uncertainty may require high-end scenarios in terms of sea-level rise (Hinkel et al., 2019), thus are in need of knowledge about high-end flood risk estimates. We therefore suggest to rather use higher percentile event durations as compared to the averaged synthetic surge shape as used in Kiesel et al. (2023). Of course, this does not mean that higher percentile events may match any future extreme event better than average surge shapes.

We have now rewritten the second half of section 2.1 in response to R2s comment and hope it is now clearer. Please find the revised text below:

*Further comparisons with the analyses of* Kiesel et al. (2023) *showed that while the peak of the 200-year return water levels by chance broadly matched the October 23 surge (Figure 1, Table 1), discrepancies existed in the temporal evolution of the synthetic surges compared to the real-world examples. Excluding the influences of short surface waves, storm tide hydrographs are a function of the mean sea level, astronomical tide and storm surge* (Lewis et al., 2011; Pugh, 1996). *Tides can be excluded as a cause for the observed differences in hydrographs, as the Baltic Sea is characterized by a microtidal regime. In addition, short surface waves were not considered in* Kiesel et al. (2023). *Differences in mean sea level might have affected the hydrographs only in terms of peak water levels, considering that the simulated 200-year design surges were detrended for sea level rise. Consequently, the differences in the shapes of the hydrographs can only originate from differences in storm characteristics.*

*The constructed design hydrographs were derived from a coastal ocean model, which covers the western Baltic Sea, using hindcast model runs (1961-2018) for each location depicted in Figure 1. Only those surges were taken into account, where the peak water level was higher than 1 m above mean sea level. Ultimately, the remaining surges were averaged in their temporal evolution* (Kiesel et al., 2023). *While the constructed surge hydrographs of* Kiesel et al. (2023) *align well with the October 23 observations within protected lagoons (Figure 2j-o), locations at the open coast show differences in the onset of event. The rise of water levels during the actual storm surge of October 23 was mostly slower than the modelled events (Figure 2a-h). This reveals an underestimation of surge duration in the constructed hydrographs at the open coast. Recent studies for the Baltic Sea cities of Lübeck and Eckernförde have demonstrated that longer surge durations at the identical peak water levels can result in larger flood extents, with variances of up to 60 %* (Höffken et al., 2020; Kupfer et al., 2024). *These studies suggest that a comparison of flood extents between the October 23 surge and the synthetic 200-year events from* Kiesel et al. (2023) *would likely reveal an underestimation in the model simulations. This has implications for coastal management, as stakeholders with a very low tolerance to uncertainty may require high-end sea-level rise scenarios* (Hinkel et al., 2019), *thus are likely in need of knowledge regarding high-end flood risk estimates in order to prepare for the worst case. Therefore, it is crucial to assess the sensitivity of flooding extents to hydrograph shapes/surge durations, as results can be highly case-specific.*

Regarding the suggestion to compare cumulative loss or damage risk estimates with observed losses, we agree that this would be a valuable addition. Unfortunately, such an analysis is beyond the scope of a brief communication and would require a dedicated study. Additionally, there are significant challenges due to the lack of detailed damage estimates in the region. For instance, the reported €200 million damage figure from the federal state of Schleswig-Holstein does not provide detailed information on specific damages within municipalities or individual floodplains. We must furthermore note that the paper we compare the October 23 surge with (Kiesel et al., 2023), has not included a flood damage assessment.

3.  The intent of the manuscript will be clearer if a section on fragility and vulnerability functions was included, especially for dikes, and how these functions are used for loss and risk estimations. This would further support the authors' arguments in Section 2. It will then be clearer that Section 2.2 highlights the need for developing accurate fragility functions for dikes based on their current deterioration state; and Section 2.3 highlights the need for developing conditional fragility functions based on the current damage state of dikes from previous events. This discussion can also improve Section 2.1 by highlighting that the fragility functions are dependent not only on peak storm levels, but are functions of the temporal evolution, and other factors such as flow velocity.

We would like to thank R2 for this important remark. We fully agree that introducing the concept of dike fragility curves will help understand our discussion around probabilistic dike failure mechanisms and will underscore why this approach is not yet widely applied in many broad-scale hydrodynamic flood risk assessments. We have now added a paragraph to the beginning of section 2.2, where we

introduce the concept of dike fragility curves and existing limitations around their application. We hope this improves the understanding of section 2.2 and 2.3. Please see the new text below:

*"In flood risk research, fragility curves are used to assess the probability of dike (or dune) failure as a consequence of a specific hydraulic loading* (Vorogushyn et al., 2009)*. However, site specific fragility curves depend on detailed information of site specific flood defense structural and foundation properties, or the uncertainty in geometrical and geotechnical dike parameters* (Simm et al., 2009)*. In addition, using two-dimensional hydrodynamic models, as is currently state-of-the-art in flood risk assessments, can yet be impractical when multiple breach and loading scenarios on large spatial scales are simulated. Among other reasons, the latter is due to high computational costs* (Simm et al., 2009)*. This reasoning might partly explain why the wider implementation of fragility curves as a probabilistic framework for studying the impacts of dike failure in broad-scale hydrodynamic flood modelling studies is currently limited. However, in most cases, the primary reason is the absence or inconsistency of geospatial data on coastal protection infrastructure."*

4. Line 139 - Minor language - …hydromorphodynamic *modelling* (Hinkel et al., 2021; Vousdoukas et al., 2018a). Ways forward *are* ultimately depend on the availability…

These mistakes have been corrected.

**References**

Gräwe, U., & Burchard, H. (2012). Storm surges in the Western Baltic Sea: the present and a possible future. *Climate Dynamics*, *39*(1), 165–183. https://doi.org/10.1007/s00382-011-1185-z

Hinkel, J., Church, J. A., Gregory, J. M., Lambert, E., Le Cozannet, G., Lowe, J., McInnes, K. L., Nicholls, R. J., van der Pol, T. D., & van de Wal, R. (2019). Meeting User Needs for Sea Level Rise Information: A Decision Analysis Perspective. *Earth's Future*, *7*(3), 320–337. https://doi.org/https://doi.org/10.1029/2018EF001071

Höffken, J., Vafeidis, A. T., MacPherson, L. R., & Dangendorf, S. (2020). Effects of the Temporal Variability of Storm Surges on Coastal Flooding. *Frontiers in Marine Science*, *7*. https://doi.org/10.3389/fmars.2020.00098

Kiesel, J., Lorenz, M., König, M., Gräwe, U., & Vafeidis, A. T. (2023). Regional assessment of extreme sea levels and associated coastal flooding along the German Baltic Sea coast. *Natural Hazards and Earth System Sciences*, *23*(9), 2961–2985. https://doi.org/10.5194/nhess-23-2961-2023

Kupfer, S., MacPherson, L. R., Hinkel, J., Arns, A., & Vafeidis, A. T. (2024). A Comprehensive Probabilistic Flood Assessment Accounting for Hydrograph Variability of ESL Events. *Journal of Geophysical Research: Oceans*, *129*(1), e2023JC019886. https://doi.org/https://doi.org/10.1029/2023JC019886

Lewis, M., Horsburgh, K., Bates, P., & Smith, R. (2011). Quantifying the Uncertainty in Future Coastal Flood Risk Estimates for the U.K. *Journal of Coastal Research*, *27*(5), 870–881. https://doi.org/10.2112/JCOASTRES-D-10-00147.1

Pugh, D. T. (1996). *Tides, Surges and Mean Sea-Levels: A Handbook for Engineers*. John Wiley & Sons.

Reusch, T. B. H., Dierking, J., Andersson, H. C., Bonsdorff, E., Carstensen, J., Casini, M., Czajkowski, M., Hasler, B., Hinsby, K., Hyytiäinen, K., Johannesson, K., Jomaa, S., Jormalainen, V., Kuosa, H.,

Kurland, S., Laikre, L., MacKenzie, B. R., Margonski, P., Melzner, F., … Zandersen, M. (2018). The Baltic Sea as a time machine for the future coastal ocean. *Science Advances*, *4*(5), eaar8195. https://doi.org/10.1126/sciadv.aar8195

Rutgersson, A., Kjellström, E., Haapala, J., Stendel, M., Danilovich, I., Drews, M., Jylhä, K., Kujala, P., Larsén, X. G., Halsnæs, K., Lehtonen, I., Luomaranta, A., Nilsson, E., Olsson, T., Särkkä, J., Tuomi, L., & Wasmund, N. (2022). Natural hazards and extreme events in the Baltic Sea region. *Earth System Dynamics*, *13*(1), 251–301. https://doi.org/10.5194/esd-13-251-2022

Simm, J., Gouldby, B., Sayers, P., Flikweert, J. J., Wersching, S., & Bramley, M. (2009). Representing fragility of flood and coastal defences: Getting into the detail. In Samuels et al. (Ed.), *Flood Risk Management - Research and Practice. Proceedings of FLOODrisk 2008*. Taylor & Francis Group.

Vorogushyn, S., Merz, B., & Apel, H. (2009). Development of dike fragility curves for piping and micro-instability breach mechanisms. *Natural Hazards and Earth System Sciences*, *9*(4), 1383–1401. https://doi.org/10.5194/nhess-9-1383-2009

Wübber, Ch., & Krauss, W. (1979). The two-dimensional seiches of the Baltic Sea. *Oceanologica Acta*, *2*(4), 435–446.

---

## Author Comment (AC2)

**Brief Communication: From modelling to reality: Flood modelling gaps highlighted by a recent severe storm surge event along the German Baltic Sea coast**

**Authors response to the reviewers**

**In order to enhance readability, our responses are written in green and in-text citations are displayed in italics.**

**Reviewer 1:**

The brief communication by Kiesel et al. provides evidence that the numerical models must be improved to simulate extreme wave inundations. This is true because the numerical models still do not accurately accommodate the temporal and spatial variation in macro-roughness during flow-structure interaction.

We thank Reviewer 1 for the positive and constructive feedback, which has helped us to improve the manuscript. In response to R1's comments, we have now clarified the role of storm characteristics for creating differences in coastal hydrographs as depicted in Figure 2 and provide explanations regarding the consideration of tides in the model and the sensitivity of the water levels with respect to return period. Please find our detailed responses below.

The communication only discusses the water levels (depths). However, water velocity is an important factor for high-energy wave impact. The mismatch between the predicted and real hydrographs would be due to the flow velocity effect but not completely attributed to the assumptions. The authors need to include a section on the significance of the flow velocity of this event and how it could affect the shape of hydrographs.

We would like to thank R1 for the interesting remark. However, we argue that differences in flow velocity are not the origin of the disagreement between the water level time series of the modelled 200-year design events and the western Baltic storm surge from October 2023 (Figure 2 in the original manuscript). This disagreement is originally caused by varying storm characteristics. The hydrographs represent coastal water level time series and the shape of the hydrograph (i.e. steep or gentle slope at the onset of the event) is a function of the storm characteristics (magnitude, duration, and storm track) and potential additional basin reactions such as seiching. In general, wind pushes water against the shore where it piles up in bays, fjords, lagoons or along beaches.

The model used to simulate the design hydrographs *(Kiesel et al., 2023)* uses wind and atmospheric pressure as forcing conditions, causing wind shear stress on the water surface and subsequent water acceleration and thus increased water velocity. This interaction, along with coastal geometry, determines the hydrograph's shape. While flow velocity is a factor, the primary driver of different hydrograph shapes at the same location is the storm's characteristics.

To address Reviewer 1's concern, we have added a paragraph to section 2.1, where we now explicitly explain how the design hydrographs were created and that differences in storm characteristics are drivers of coastal hydrograph variability (i.e. differences in storm surge durations):

*"Excluding the influences of short surface waves, storm tide hydrographs are a function of the mean sea level, astronomical tide and storm surge* (Lewis et al., 2011; Pugh, 1996). *Tides can be excluded as a cause for the observed differences in hydrographs, as the Baltic Sea is microtidal. In addition, short surface waves were not considered in* Kiesel et al. (2023). *Differences in mean sea level might have affected the hydrographs only in terms of peak water levels, considering that the simulated 200-year design surges were de-trended for sea level rise. Consequently, the differences in the shapes of the hydrographs can only originate from differences in storm characteristics. The constructed design hydrographs were derived from a coastal ocean model, which covers the western Baltic Sea, using hindcast model runs (1961-2018) for each location depicted in Figure 1. Only those surges were taken into account, where the peak water level was higher than 1 m above mean sea level. Ultimately, the remaining surges were averaged in their temporal evolution* (Kiesel et al., 2023). *"*

The authors selected a 200-year return period for simulating water levels. Was the tidal effect included in the simulation?

The tidal effect was not included in the model that simulated the hydrographs shown in Figure 2. The reason for the exclusion is that the tidal range in the Baltic Sea is very small, less than 10 cm in its southwestern parts (e.g. Gräwe & Burchard (2012)). The more important periodic oscillations that are included in our simulations are seiches (inertial oscillations caused by perturbations in wind and air pressure fields, i.e. storms, e.g. Wübber & Krauss (1979)). To clarify this in the manuscript, we have now added the following sentence to the introduction and main body of the text.

Introduction:

*"The Baltic Sea is a semi-enclosed, microtidal marginal sea in the eastern North Atlantic, which is under pressure from a multitude of anthropogenic disturbances and natural hazards* (Reusch et al., 2018; Rutgersson et al., 2022)*.*

Many Body:
*" Tides can be excluded as a cause for the observed differences in hydrographs, as the Baltic Sea is microtidal."*

What is the sensitivity of the water level to the return period?

The sensitivity of the water level with respect to return period is shown for one example GEV distribution for the station Kiel, Germany (Figure 1, taken from Kiesel et al. (2023)). The difference in peak water level for a 30- and 200-year event amounts to approximately 30 cm (see also Table 4 in Kiesel et al. (2023)).

[Figure]

Figure 1: Peak water level and associated return period for the station Kiel-Holtenau, Germany (Kiesel et al., 2023).

**References**

Gräwe, U., & Burchard, H. (2012). Storm surges in the Western Baltic Sea: the present and a possible future. *Climate Dynamics*, *39*(1), 165–183. https://doi.org/10.1007/s00382-011-1185-z

Kiesel, J., Lorenz, M., König, M., Gräwe, U., & Vafeidis, A. T. (2023). Regional assessment of extreme sea levels and associated coastal flooding along the German Baltic Sea coast. *Natural Hazards and Earth System Sciences*, *23*(9), 2961–2985. https://doi.org/10.5194/nhess-23-2961-2023

Lewis, M., Horsburgh, K., Bates, P., & Smith, R. (2011). Quantifying the Uncertainty in Future Coastal Flood Risk Estimates for the U.K. *Journal of Coastal Research*, *27*(5), 870–881. https://doi.org/10.2112/JCOASTRES-D-10-00147.1

Pugh, D. T. (1996). *Tides, Surges and Mean Sea-Levels: A Handbook for Engineers*. John Wiley & Sons.

Reusch, T. B. H., Dierking, J., Andersson, H. C., Bonsdorff, E., Carstensen, J., Casini, M., Czajkowski, M., Hasler, B., Hinsby, K., Hyytiäinen, K., Johannesson, K., Jomaa, S., Jormalainen, V., Kuosa, H., Kurland, S., Laikre, L., MacKenzie, B. R., Margonski, P., Melzner, F., … Zandersen, M. (2018). The Baltic Sea as a time machine for the future coastal ocean. *Science Advances*, *4*(5), eaar8195. https://doi.org/10.1126/sciadv.aar8195

Rutgersson, A., Kjellström, E., Haapala, J., Stendel, M., Danilovich, I., Drews, M., Jylhä, K., Kujala, P., Larsén, X. G., Halsnæs, K., Lehtonen, I., Luomaranta, A., Nilsson, E., Olsson, T., Särkkä, J., Tuomi, L., & Wasmund, N. (2022). Natural hazards and extreme events in the Baltic Sea region. *Earth System Dynamics*, *13*(1), 251–301. https://doi.org/10.5194/esd-13-251-2022

Wübber, Ch., & Krauss, W. (1979). The two-dimensional seiches of the Baltic Sea. *Oceanologica Acta*, *2*(4), 435–446.